# Little-Parks like oscillations in lightly doped cuprate superconductors

Menghan Liao[1], Yuying Zhu[2], Shuxu Hu[1], Ruidan Zhong[3], John Schneeloch[3,4], Genda Gu[3], Ding Zhang [1,2,5,6 ✉] & Qi-Kun Xue[1,2,5,7 ✉]

Understanding the rich and competing electronic orders in cuprate superconductors may provide important insight into the mechanism of high-temperature superconductivity. Here, by measuring $Bi_2Sr_2CaCu_2O_{8+x}$ in the extremely underdoped regime, we obtain evidence for a distinct type of ordering, which manifests itself as resistance oscillations at low magnetic fields ($\leq 10$ T) and at temperatures around the superconducting transition. By tuning the doping level $p$ continuously, we reveal that these low-field oscillations occur only when $p < 0.1$. The oscillation amplitude increases with decreasing $p$ but the oscillation period stays almost constant. We show that these low-field oscillations can be well described by assuming a periodic superconducting structure with a mesh size of about 50 nm. Such a charge order, which is distinctly different from the well-established charge density wave and pair density wave, seems to be an unexpected piece of the puzzle on the correlated physics in cuprates.

[1] State Key Laboratory of Low Dimensional Quantum Physics and Department of Physics, Tsinghua University, 100084 Beijing, China. [2] Beijing Academy of Quantum Information Sciences, 100193 Beijing, China. [3] Condensed Matter Physics and Materials Science Department, Brookhaven National Laboratory, Upton, NY 11973, USA. [4] Department of Physics and Astronomy, Stony Brook University, Stony Brook, NY 11794, USA. [5] Frontier Science Center for Quantum Information, 100084 Beijing, China. [6] RIKEN Center for Emergent Matter Science (CEMS), Wako, Saitama 351-0198, Japan. [7] Southern University of Science and Technology, 518055 Shenzhen, China. ✉email: dingzhang@mail.tsinghua.edu.cn; qkxue@mail.tsinghua.edu.cn

A fundamental property of superconductivity is the fluxoid quantization, which can manifest itself in a super-conducting hollow cylinder through a periodically modulated superconducting transition temperature ($T_c$) as a function of magnetic field—the so-called Little-Parks effect[1,2]. It stems from the modulated kinetic energy of the supercurrent as a joint effect of the discrete jumps in the fluxoid winding number and the continuously increasing external magnetic field. Experimentally, Little and Parks observed oscillations in the magneto-resistance at a fixed temperature close to $T_c$[1], because a modulated $T_c$ could be directly translated to the oscillating resistance via: $\Delta R = \Delta T_c \times dR/dT$. Lately, such a fundamental type of resistance oscillations has been employed not only for the demonstration of pairing in more exotic situations such as the superconductor-insulator transition regime[3,4] but also for addressing the unconventional pairing symmetry[5]. In most cases, the material under investigation needs to be nanopatterned into a ring or a periodic network[6–9]. Observing this type of oscillations in superconductors without artificial patterning often suggests highly non-trivial quantum physics at play, such as the presence of chiral edge state in $MoTe_2$[10] or a spontaneously emerged periodic order in $TiSe_2$[11,12]. The latter phenomenon was attributed to the periodic domains of commensurate charge density wave (CCDW) separated by incommensurate charge density wave (ICDW). Understanding the charge orders is also of high interest due to their omnipresence among high-temperature cuprate superconductors[13,14]. So far, charge ordering in cuprates was primarily addressed by sophisticated techniques such as resonant X-ray scattering (RXS) or scanning tunneling microscopy (STM). Here we report evidence for charge ordering, although of a distinct type, in a high temperature superconductor—$Bi_2Sr_2CaCu_2O_{8+x}$ (Bi-2212) through transport measurements. Our work provides a fresh perspective at the interplay between superconductivity and charge ordering in the underdoped regime.

## Results

**Characterization of lightly doped samples**. We investigate transport properties of Bi-2212 over a wide doping range by further optimizing the technique we developed previously[15], which includes fabrication of ultrathin Bi-2212 samples and ionic solid gating. Figure 1a shows one of our samples (S1)—a mechanically exfoliated flake of Bi-2212 (highlighted in blue, ~50 nm thick, $80 \times 220\,\mu m^2$ lateral size). By applying positive gate voltages (1–2 V) to the backside of the substrate—a solid ion conductor, lithium ions can intercalate into Bi-2212 (inset of Fig. 1a). We gate S1 for a certain period of time (right panels of Fig. 1b) and collect temperature ($T$) dependent resistances after each step of gating. Figure 1b shows that the normal-state resistance increases and the zero-field superconducting transition temperature $T_{c0}$ drops consecutively, demonstrating the tunability from the nearly optimal doped state ($p = 0.16$) to the extremely underdoped region ($p \sim 0.05$).

At each fixed $p$, we measure the magneto-resistance. Figure 1c gives exemplary traces of S1 after the fifth gating (S1-#5), corresponding to an underdoped state with $T_{c0} = 13.7$ K ($p = 0.06$). A finite resistance sets in at a fairly small magnetic field, which reflects the melting of a vortex solid into the liquid phase in high-temperature superconductors[16–19]. In comparison, conventional low-temperature superconductors usually exhibit a clear transition from zero-resistance to finite resistance at the upper critical field—$B_{c2}$. Despite the complication brought by the vortex liquid phase, we extrapolate the melting field—$B_{vs}(T)$ (demarcated in Fig. 1d as a white curve) to absolute zero to estimate a lower bound of $B_{c2,min}$[19] (Supplementary Note 1). The superconducting coherence length $\xi_0$ can be determined via $\xi_0 = \sqrt{\Phi_0/(2\pi B_{c2,min})}$,

where $\Phi_0 = h/2e$ is the flux quantum. For S1-#5, we obtain $\xi_0 = 4.6$ nm, in agreement with values obtained for underdoped $YBa_2Cu_3O_{6+x}$ (YBCO) or $Bi_2Sr_2CuO_{6+x}$ (Bi-2201)[18,19]. (The estimated values of $\xi_0$ at other doping levels are shown in Supplementary Fig. 2.) The temperature dependence of $B_{c2}(T)$ is schematically drawn in Fig. 1d by connecting $(B, T) = (0, T_{c0})$ to $(B_{c2,min}, 0)$ as done in Grissonnanche et al.[19]. with a quadratic curve.

**Magneto-resistance oscillations**. On top of the increasing trend of the magneto-resistances, we observe regular dips, which occur repeatedly in multiple samples, including a one unit-cell (1-UC) thick Bi-2212 without lithium intercalation (Figs. 1c and 2 and Supplementary Figs. 3 and 4). Samples were measured in two cryogenic systems and we further excluded possible extrinsic contributions by sweeping the magnetic field with different rates, in different directions, in different cool-downs, and applying different excitation currents (Supplementary Fig. 5). Figure 1c shows that the oscillations occur at the same $B$-field positions symmetrically around $B = 0$ T for different scans at various temperatures. Figure 1d shows data in a finer temperature step, where we subtract each curve with a smoothed background (Supplementary Note 5). Vertical zebra stripes appear below the $B_{vs} - T$ curve. They always occur far below $B_{c2}$, such that the contribution from the normal state remains minimal. The stripes vanish when Bi-2212 plunges deep into the superconducting state. In Fig. 1e, we carry out fast Fourier transformation (FFT) of the traces at different temperatures by assuming $B$-periodic oscillations. A sharp peak occurs below 7 K and becomes most prominent at around 5 K, which corresponds to the trace in Fig. 1c (dark green) showing almost linear magnetoresistance. The FFT peak position stays the same—1.11 $T^{-1}$ (vertical dashed line) at different temperatures. It corresponds to a period in $\triangle B$ of 0.9 T.

We further show that the oscillations are of two-dimensional (2D) nature. Here, we measure sample S2 at a similar doping to S1-#5. We employ a setup where the sample can be rotated within an applied magnetic field. The measurement temperature is chosen such that the general behavior of the magneto-resistance is linear, in order to better appreciate the oscillating features. Figure 1f plots the magnetoresistance of S2 as a function of the perpendicular component of the magnetic field. Clearly, the peaks and dips in traces obtained at different rotation angles occur at the same perpendicular magnetic fields, although the corresponding in-plane magnetic fields are sharply different.

In addition, samples S1 to S4 show multiple superconducting transitions, indicating possible inhomogeneity in doping. However, we observe pronounced resistance oscillations only in the extremely underdoped regime, in which the temperature dependent resistance shows predominantly a single transition. By contrast, in the gated state where the multi-step transition is most pronounced (gated state #1 to #3 of S1, for example), we observe no magneto-resistance oscillations. For sample S5 and S6 shown in the supplementary information (Supplementary Figs. 3 and 4), under-doping is achieved by natural loss of oxygen. Apart from the superconducting transition at the corresponding $T_{c0}$ of the underdoped situation, a small kink at around 87 K can be identified. However, this kink clearly originates from the nearby regions (thicker flakes, for example) with the initial optimal doping. The underdoped regions in S5 and S6 essentially possess only a single superconducting transition. Still, we observe pronounced resistance oscillations. The resistance oscillations are therefore not correlated with the apparent multi-step transition.

**Doping dependence**. Facilitated by our in-situ gating techniques, we vary the doping level in a single sample and investigate the

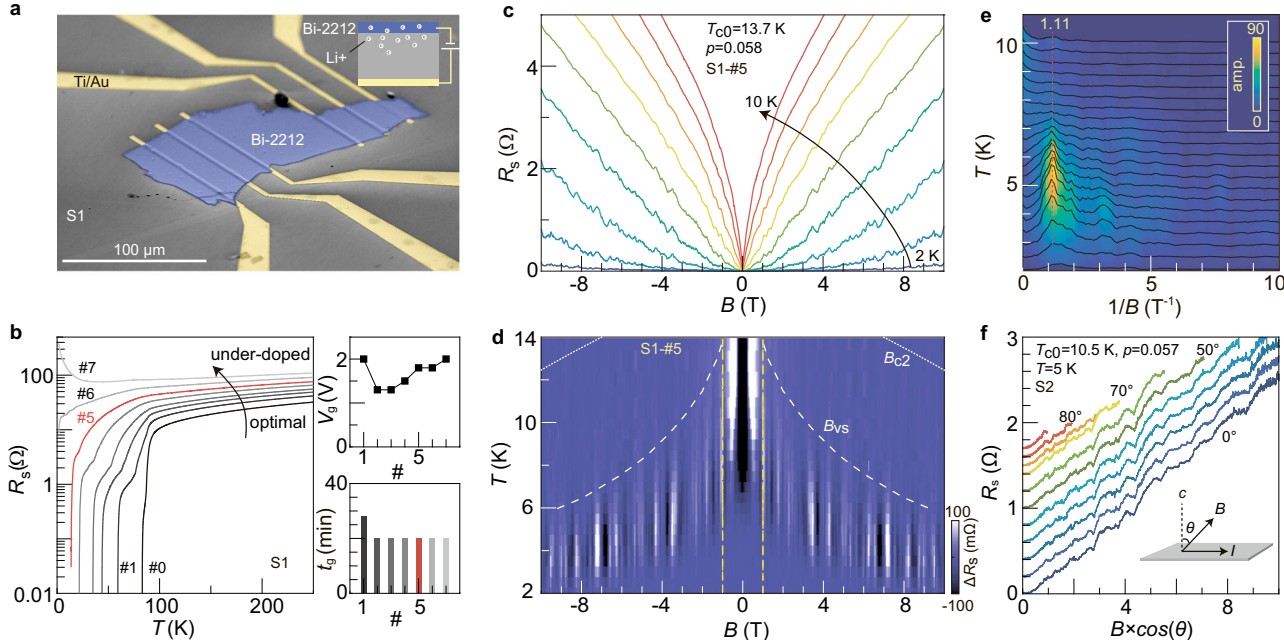

**Fig. 1 Resistance oscillations in underdoped Bi-2212. a** False-color scanning electron microscopy image of sample S1. The Bi-2212 flake (electrode) is highlighted in blue (yellow). The substrate is a solid ion conductor containing lithium ions. The inset illustrates the gating configuration. **b** Main panel: temperature dependent resistances of S1 at sequentially gated states. Right panels: gating voltages ($V_g$) and estimated gating time periods ($t_g$) for each sequence. $t_g$ includes the complete course of gating at three stages: warming up from 280 K to 300 K; staying at 300 K; cooling down from 300 K to 280 K. $V_g$ is applied throughout the above-mentioned course of time. **c** Sheet resistance of S1 in an underdoped state (S1-#5) as a function of magnetic field at a selected set of temperatures (from 2 to 10 K in steps of 1 K). We estimate the doping level $p$ by using the experimentally measured $T_{c0}$ and the formula: $T_{c0} = T_{c,max}[1 - 82.6(p - 0.16)^2]$. **d** Color plot of the background subtracted resistance $\triangle R$ of S1-#5. Bright and dark stripes in the magnetic field range from $-1$ to 1T (dashed vertical lines) are artefacts from over-smoothening. Data outside this region is not affected. White dashed curve indicates the line where the resistance reaches 10% of the normal state resistance (zero-field resistance value at 87 K). It represents the phase boundary between the vortex solid (vs) and the vortex liquid. White dotted curve is a guide to the eye, illustrating the **e** expected temperature dependence of the upper critical fields. **e** Fast Fourier transform (FFT) of the resistance traces at different temperatures in **d**. The dotted line with the number marks the peak position in FFT. **f** Angular dependent study of the magneto-resistance of S2 as a function of the perpendicular magnetic field. The angle $\theta$ is between the total magnetic field and the normal to the sample plane (inset). It increases from 0° to 70° in steps of 10° and from 75° to 85° in steps of 5°. Curves are vertically offset for clarity.

evolution of resistance oscillations. Figure 2 shows data sets of sample S3. We select the representative results at 6 doping levels out of the total 10 superconducting states we realized. They are arranged from left to right with decreasing $p$, which follows the gating sequence. Figure 2a shows the color-coded magneto-resistance. In each panel, the purple areas represent the normal state and the dark blue regions indicate the superconducting regime. The white color corresponds to the resistance value that is half of the normal state resistance, indicating the superconducting transition region. Figure 2b shows the background subtracted data $\triangle R(B, T)$. The zebra stripes are prominent in Fig. 2b2–b6 ($p \leq 0.084$) but are missing in Fig. 2b1 ($p = 0.1$). These stripes always appear in the superconducting transition region (slightly below the white bands in figures of the top row of Fig. 2), reaffirming their intimate link to superconductivity. The oscillation amplitude increases monotonically with decreasing doping as indicated by the increasing scales of the color bars. In Fig. 2c, we show the FFT results at different doping levels. The vertical dotted lines mark the peak positions, which seem to show little doping dependence. Their values are consistent with that obtained in sample S1. In Fig. 3, we summarize the experimentally investigated temperature points and doping levels for five samples. We use filled symbols to indicate that resistance oscillations (RO), such as those shown in Fig. 1c, d, f and Fig. 2b2–b6, are experimentally observed. The filled points cluster in the lightly doped regime ($p \leq 0.08$) of the superconducting dome. It indicates that RO are reproducible across different samples, as long as the samples are tuned into the extremely underdoped regime.

## Discussion

First, we exclude quantum oscillations of the normal state as the origin for the observed oscillations. As alluded to before, the resistance oscillations only occur at magnetic fields below the phase boundary between the vortex solid and vortex liquid. These magnetic fields are far below the upper critical field, such that the contribution from the normal state can be neglected. Furthermore, gating our sample into the non-superconducting state (the left-most column of data points in Fig. 3), the resistance oscillations disappear in the experimentally accessible temperature range (down to 1.6 K). It indicates that the resistance oscillations are closely related to the superconducting state.

Studying quantum oscillations in cuprates, instead, often requires the combination of ultrahigh-quality single crystals [typically YBa$_2$Cu$_3$O$_{6+x}$ (YBCO) or Tl$_2$Ba$_2$CuO$_{6+x}$ (TBCO)] and high magnetic field facilities. The necessity of a powerful magnet is not only because of large upper critical fields in cuprates. It also stems from the criterion for observing quantum oscillations:[20,21] charge carriers should be able to complete at least one cyclotron orbit before scattering, i.e. $\omega_c \tau \geq 1$, where $\omega_c$ is the cyclotron frequency and $\tau$ is the scattering time. This requirement can be equivalently expressed by the product of carrier mobility ($\mu$) and magnetic field (in SI units): $\mu B \geq 1$, because $\mu B = e\tau/m^*B = \omega_c\tau$, where $m^*$ is the effective mass. For high quality YBCO and TBCO crystals, $\mu$ is on the order of 0.001–0.01 m$^2$/V s (10 to 100 cm$^2$/V s)[20,21], allowing the observation of quantum oscillations at $B \sim 50$–100 T. In comparison, we estimate that the mobility of Bi-2212 is ~1 cm$^2$/V s[15], such that quantum oscillations are

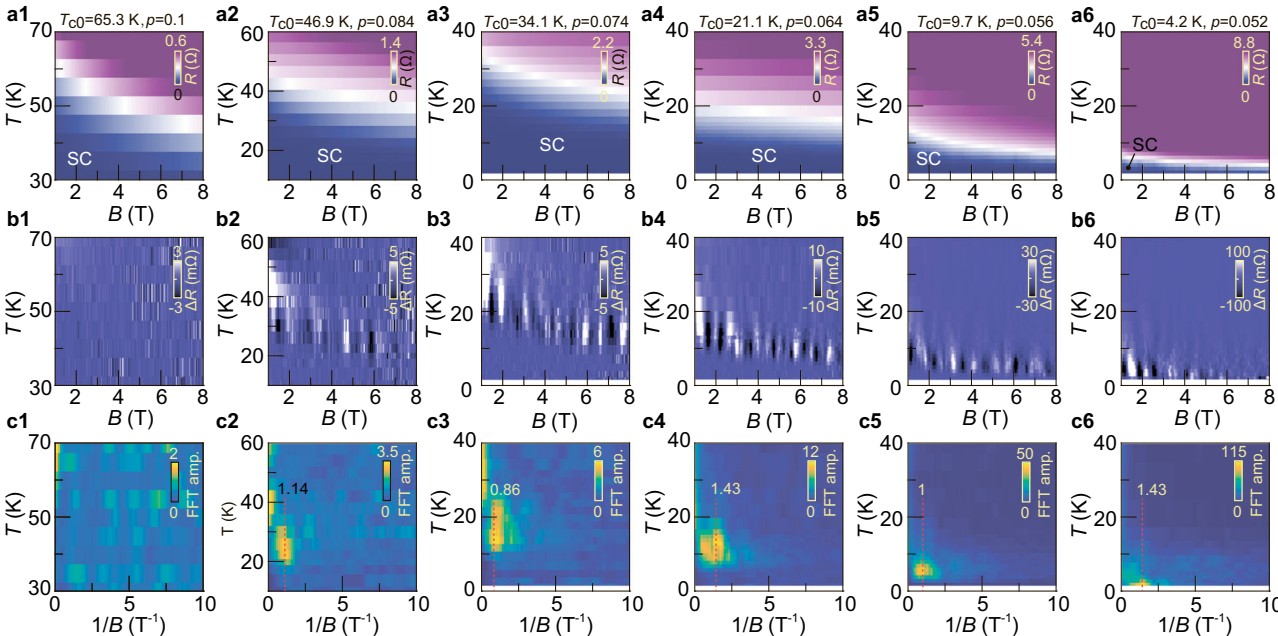

**Fig. 2 Doping dependence of the magneto resistance and the oscillations. a1–a6** Color-coded resistance of sample S3 as a function of magnetic field and temperature $R(B, T)$. The white stripe demarcates the boundary between the superconducting (SC) region and the normal state. **b1–b6** Background subtracted resistance as a function of magnetic field and temperature: $\Delta R(B, T)$. The oscillation amplitude increases with decreasing doping. **c1–c6** FFT of $\Delta R(B)$ at different temperatures. Vertical dotted lines with numbers indicate the peak positions. Each column of this figure corresponds to the same doping level indicated on the top.

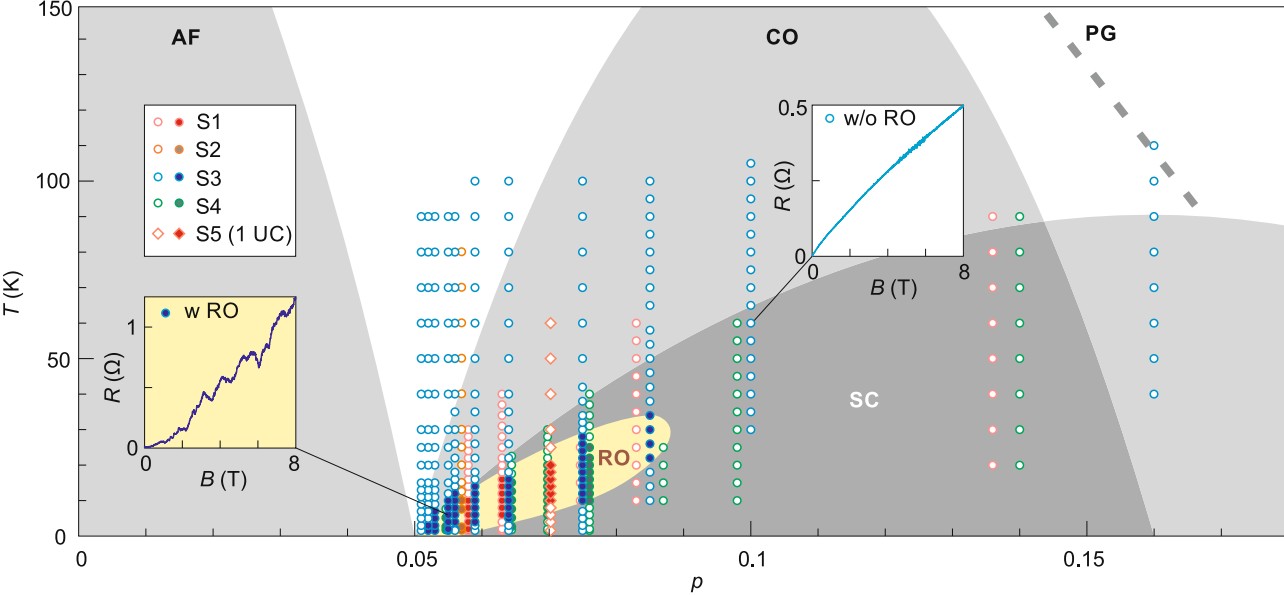

**Fig. 3 Phase diagram of the Little-Parks like resistance oscillations in Bi-2212.** Each symbol represents a fixed set of temperature and doping at which the magnetoresistance measurement was carried out. Filled (empty) symbols indicate that resistance oscillations—RO—were present (absent). Exemplary traces are shown in the inset boxes. Overlapping data points at $p=0.055$, $0.064$, $0.07$ and $0.075$ are slightly shifted horizontally for better presentation. In the low temperature regime where resistance oscillations occurred, the actual measurements were taken with denser sets of temperature points (with steps of $\Delta T = 0.5$ K) than represented here ($\Delta T = 2$ K). Gray shadows and dashed curve indicate the typical phases: anti-ferromagnetism (AF), charge order (CO), pseudo-gap (PG), and superconductivity (SC). Here data points marked as S1 and S4 are from the same flake but with two different pairs of contacts in two separate regions. S5 is a 1-UC Bi-2212 flake that is in the underdoped regime after exfoliation. Yellow shadow highlights the region where RO can be experimentally observed.

expected to occur at $B>10^4$ T. Any oscillations observed below 10 T cannot stem from quantum oscillations of the normal state.

Second, our results differ from the low-$B$ oscillations of mesoscopic origin as reported previously. Table 1 summarizes the comparison. Resistance oscillations were observed in nanobridges or nanowires of a variety of superconductors[22–25]. They were attributed to the rearrangement of vortex lattice in the bridge/wire or formation of multiply connected loops[26]. These mesoscopic effects play a major role for samples with a width that is comparable to $\xi_0$. By contrast, our sample has a width on the

**Table 1 Comparison between the resistance oscillations of this work and the previously reported low-field oscillations in transport.**

| Previous studies | | This study |
|---|---|---|
| Superconducting nano-bridge[22-25] | Sample width $\sim \xi$ | Sample width $\gg \xi$ |
| Superconducting strip[27] | Isotropic | Anisotropic (2D) |
| | $\Delta B \sim 0.1$ T | $\Delta B \sim 1$ T |
| Nernst effect in Bi-2212[28] | OP-Bi2212: yes | OP-Bi2212: no |
| | UD-Bi2212: no | UD-Bi2212: yes |
| Little-Parks oscillations in TiSe$_2$[11,12] | $T \sim 1$ K, $B < 0.6$ T | $T \sim 10$ K |

order of tens of micrometers, i.e. $10^5 \xi_0$. The mesoscopic effect should be substantially suppressed in our case. In fact, we find that narrowing the width down to 1 μm neither enhances the resistance oscillations nor changes the period (Supplementary Note 3), further excluding the possible involvement of mesoscopic effects. Recently, anomalous resistance oscillations with pronounced periodicity were reported in various superconducting films[27]. However, the reported oscillations showed isotropic magnetic field dependence, different from the behavior seen in Fig. 1f. Their period is also one order of magnitude smaller.

Apart from resistance oscillations, Nernst effect, which is particularly sensitive to vortex motions, revealed oscillations in Bi-2212[28]. There, oscillations emerged at the beginning of the rising slope of the Nernst signal, where the vortex solid just started to melt into the liquid phase. It was thus speculated that, around the solid–liquid phase boundary, the vortex lattice may still remain in some regions. Adding new vortices causes plastic deformation such that the Nernst signal fluctuates. At still higher magnetic field—deep in the vortex liquid phase, no more plastic deformation would occur, and the Nernst signal showed a monotonic increase. However, the oscillations caused by the plastic flow of vortices are not expected to occur at the same perpendicular fields with different in-plane magnetic fields, as we observed in Fig. 1f. The energetics of vortices change drastically once an in-plane magnetic field is introduced, because additional Josephson vortices emerge[29]. Additionally, the oscillations in Nernst signal only occurred at optimal doping and vanished in the underdoped case. It indicates that the effect of plastic flow of vortices is suppressed in the underdoped regime. By contrast, none of our samples at optimal doping showed oscillations in magneto-resistance. Resistance oscillations only occur deep in the underdoped regime. Therefore, electrical resistance and thermoelectric Nernst measurements probably probe oscillations of different origins.

We now consider the relation between our results and the Little-Parks oscillations. Our results are reminiscent to the intriguing phenomenon recently observed in ionic liquid gated TiSe$_2$[11] and lithium intercalated TiSe$_2$[12], but at temperature and magnetic field that are both over one order of magnitude higher. In TiSe$_2$, it was proposed that a periodic structure emerged: CCDW patches with ICDW stripes at the domain boundaries. At low temperatures, the ICDW regions became superconducting while CCDW regions remained non-superconducting. Cooper pairs going around the superconducting periodic mesh can experience the Little-Parks effect. The superconducting transition temperature oscillates as a function of magnetic field with a period of $\Delta B = \Phi_0/a^2$, where $a$ is the spatial periodicity. At a fixed $T$ close to the unmodulated $T_c$, the oscillating part of $T_c$ in turn gives rise to resistance oscillations with the same period. In what follows, we assume that a similar periodic structure occurs spontaneously in Bi-2212, as schematically drawn in Fig. 4a. We

then test the corresponding theoretical models and discuss the implications. Figure 4a summarizes the extracted periodicity $a$ from all the samples we investigated. The periodic structures are very similar across samples with different thicknesses and sizes, reaffirming an intrinsic origin. It also indicates that this effect is essentially representative of bulk crystals, although technically thin flakes can be more easily tuned in the doping level.

Apart from the oscillating period, we carry out a quantitative analysis of the oscillating amplitude. At each $T$ and $p$, we extract $\Delta R$ from the difference between the neighboring local minimum and maximum in the resistance trace as a function of $B$, after subtracting the smoothed background. Figure 4c plots the typical results from sample S3 as a function of $T$. In comparison, we theoretically estimate the oscillating amplitude expected for the Little-Parks effect. For a square array with a side length of $a$ (Fig. 4a), the Little-Parks effect gives rise to an oscillation depth $\Delta T_c$:[30] $\Delta T_{c,L-P}/T_c = \pi^2/16(\xi_0/a)^2$. The oscillation amplitude in resistance $\Delta R_{L-P}$ can be then evaluated via $\Delta R_{L-P} = \Delta T_{c,L-P}(dR/dT)$. By using the experimentally determined $\xi_0$, $a$ and $dR/dT$, we obtain $\Delta R_{L-P}$ as a function of $T$, which is plotted as dashed curves in Fig. 4c. Clearly, $\Delta R_{L-P}$ is much weaker than the experimentally observed oscillations, albeit with similar temperature dependence.

In order to quantitatively describe the resistance oscillations, we have to take into account the vortex effect in a periodic superconducting array. In cuprate superconductors, vortices play a much important role than those in low temperature superconductors do[29]. Thermally excited vortices can cross the superconducting array. This process causes fluxoid transition and modulates $T_c$ in a similar fashion to that of Little-Parks effect (see inset of Fig. 4a). Such a vortex effect was demonstrated to contribute predominantly to the resistance oscillations in periodically patterned La$_{2-x}$Sr$_x$CuO$_4$ films[8]. In this scenario, the oscillating amplitude follows the formula:[8]

$$\Delta R = R_n \left(2E_1/k_B T\right)^2 \frac{I_1\left[(E_v + E_1)/2k_B T\right]}{\left[I_0\left[(E_v + E_1)/2k_B T\right]\right]^3} \quad (1)$$

Here $R_n$ is the normal state resistance, $I_0$ and $I_1$ are the zero- and first-order modified Bessel function of the first kind, respectively. The two energy quantities are: $E_1 = \Phi_0^2/8\pi\mu_0\Lambda(T) \cdot w/a$ and $E_v = \Phi_0^2/2\pi\mu_0\Lambda(T) \cdot ln(2w/\pi\xi(T))$, where $\mu_0$ is the vacuum permeability, $w$ is the separation between the squares (inset of Fig. 4a). $\Lambda(T)$ is the Pearl penetration depth: $2\lambda(T)^2/d$, where $d$ is the superconducting thickness. The temperature dependent penetration depth and coherence length are: $\lambda(T) = \lambda_0/\sqrt{1 - (T/T_c)^2}$, $\xi(T) = 0.74\xi_0/\sqrt{1 - T/T_c}$.

As shown in Fig. 4c, d as solid curves, our data can be nicely fit by Eq. (1). Here we plug in the estimated $\xi_0$, $a$ and $T_c$ and use $R_n$, $\lambda_0$ and $w$ as three fitting parameters (Supplementary Note 6 and Supplementary Table 1). Figure 4b and Supplementary Fig. 7 summarize the extracted values of $\lambda_0$ from different samples. Together, they suggest a clearly increasing trend with decreasing doping. This behavior is understandable, because the superfluid density, which is inversely proportional to the square of the penetration depth, decreases with reduced doping in the underdoped regime. The extracted values of $\lambda$ also follow the doping dependence extrapolated from previous experimental results on Bi-2212 superconductors[31,32] (see Supplementary Fig. 7). The consistent values and doping dependence suggest that our speculation of a periodic structure captures important physics of the observed phenomenon.

Our results indicate that there exists a distinct phase in the lightly doped cuprate superconductors with large spatial periodicity of ~50 nm (hereafter referred to as P-50). The surprisingly large periodic structure has not been reported before. RXS and

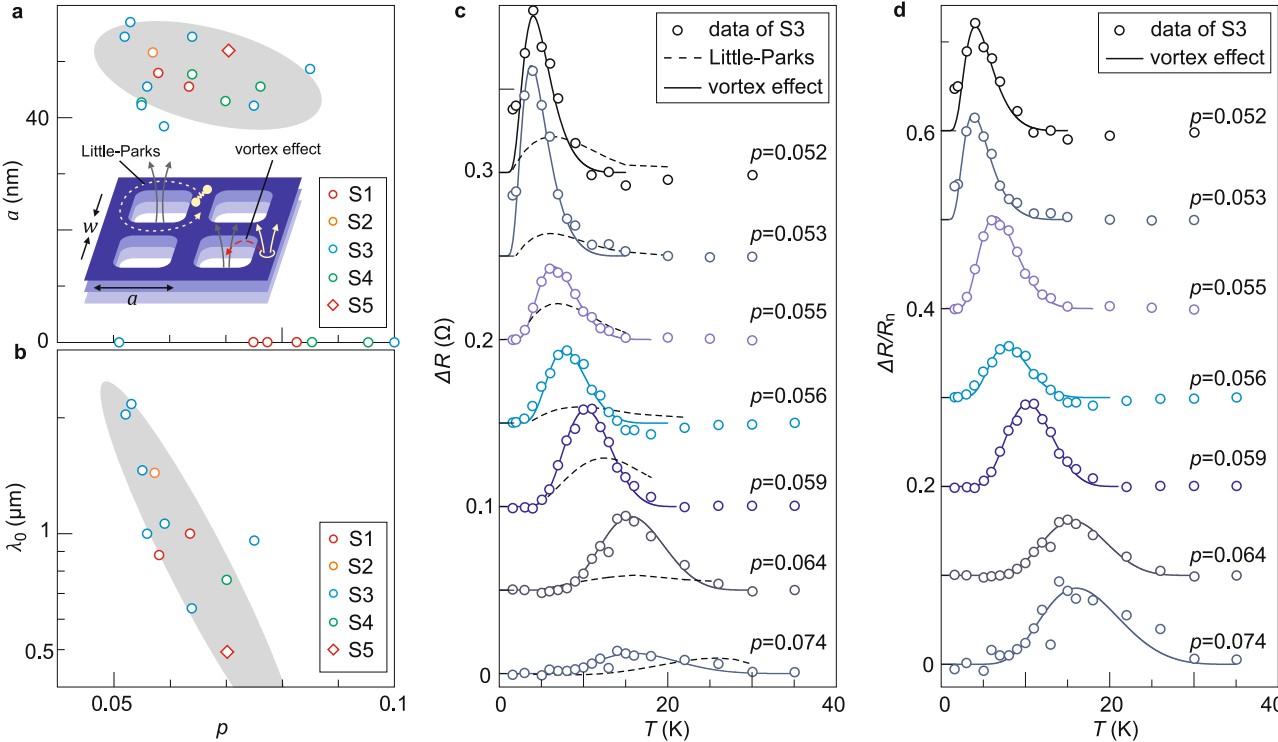

**Fig. 4 Quantitative analysis of the Little-Parks like resistance oscillations. a** Extracted spatial period as a function of doping for different samples. Inset schematically illustrates the Little-Parks effect and the vortex effect in a periodic superconducting array. The colored mesh indicates the superconducting region. The empty squares indicate the non-superconducting region. **b** Penetration depth extracted by fitting the temperature dependent oscillation amplitude. **c** oscillation amplitude as a function of temperature $\Delta R(T)$ at different doping levels for sample S3. They are obtained by calculating the absolute difference between the local and nearest-neighboring maximum and minimum in the $\Delta R(B)$ traces ($B \in^{1,4}$ T) at each $p$ and $T$. Dashed curves are the expected oscillation amplitudes according to the Little-Parks effect. Solid curves are the theoretical fits by taking into account the vortex effect [Eq. (1)]. Data points and curves are vertically offset by 0.05 Ω for clarity. **d** same data sets as shown in **c** but normalized by the normal state resistance $R_n$. They are vertically offset by 0.1 for clarity.

STM experiments identified charge ordering (CO) with a small period of about $4a_0$[13,14]. As shown in Fig. 3, CO spans in a wide doping range of the phase diagram. By contrast, P-50 here emerges only in the extremely underdoped regime $p \leq 0.08$. Despite the sharp differences, there could be an intimate link between P-50 and CO, in a similar fashion to those periodic structures observed in TaS₂[33] and speculated to exist in TiSe₂[11,12]. In these transition metal dichalcogenides, CCDW and ICDW phases are identified in the phase diagram as a function of doping or pressure. Individually, CCDW and ICDW have small periods, but a large periodic structure emerges in the regime where they coexist. Around $p = 0.1$, CO is commensurate with a period of $4a_0$. By decreasing $p$, previous experiments observed a drop in the spatial period by ~15% when $p$ changes from $p = 0.1$ to $p = 0.05$, suggesting that CO becomes incommensurate[13,14]. We conjecture that such an evolution may not be uniform spatially. CO with different periods may coexist at $p < 0.1$. If there exist predominantly regions with two periods, for example: $(4 - \delta - \frac{\sigma}{2})a_0$ and $(4 - \delta + \frac{\sigma}{2})a_0$, where $\delta < 0.5$ and $\sigma \ll 0.1$, a large superstructure may appear with a period at around $(4 - \delta)a_0/\sigma$. Here $\sigma$ only need to be as small as 0.03 to account for the 50-nm period. The weak doping dependence of $a$ (see Fig. 4b) can also be explained. It suggests that with increasing $\delta$ at lower $p$ the spread of $\sigma$ becomes smaller. Although the above scenario remains speculative, a recent finding indeed showed that CDW in cuprates tend to be locally commensurate but globally incommensurate[34], suggesting the coexistence and competition between them.

In summary, we study transport properties of Bi-2212 thin flakes in the underdoped regime and observe peculiar resistance

oscillations at low magnetic fields. Samples with different thicknesses and sizes reproducibly show low-$B$ resistance oscillations in the lightly doped regime with consistent oscillating periods. It indicates that there exists a spontaneously emerged periodic structure in the underdoped Bi-2212 ($p \leq 0.08$) with a size of about 50 nm. We speculate that such a large periodic structure may emerge as a competition between the commensurate and incommensurate charge ordering. Our work not only highlights the rich and enticing physics in extremely underdoped high temperature superconductors but also demonstrates a facile technique to access this less-explored regime. Future experimental work may be devoted to unveiling the large periodic structure in real space by scanning probe microscopy.

## Methods

Bi-2212 thin flakes were fabricated in a glovebox with Ar atmosphere (H₂O < 0.1 ppm, O₂ < 0.1 ppm). Mechanical exfoliation was carried out by using standard scotch tape technique. Thin flakes were first exfoliated onto Gel-Films and then dry-transferred onto solid ion conductors or SiO₂/Si substrates[15]. For sample S1, S4, and S5, we pre-patterned electrodes (5 nm Ti/30 nm Au) and then placed the Bi-2212 flake on top of the electrodes. For sample S2 and S3, we deposited thin graphite flakes as bottom contacts and then placed BSCCO on top. For the 1UC-Bi-2212 flake, we further capped it with a thin flake of hexagonal boron nitride (hBN) as a protection layer. Samples were then taken out for wire-bonding and subsequent measurements. Silver paste was applied to the backside of the solid ion conductor as the back-gating electrode. The time period for air exposure was kept as short as 20 min.

Transport measurements were carried out in two closed-cycle cryogenic systems (Oxford Instruments TelatronPT: 1.55–300 K, 9/12 T) with three different inserts. The electrical resistance was measured in a standard four-terminal configuration by using either the AC lock-in technique (S1, S4, and S5, excitation current: 1 μA) or a combination of DC current source and voltmeter (Keithley 6221 and 2182 A) in the

delta mode (S2 and S3, excitation current: 10 μA). The DC gate voltage was applied with a source measure unit (Keithley 2400).

## Data availability

All data needed to evaluate the conclusions are available in the main text or the supplementary information.

## Code availability

The computer code used for data analysis is available upon request from the corresponding author.

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

## Acknowledgements

We thank Hong-Yi Xie, Haiwen Liu, Yayu Wang, Hong Yao, and Fa Wang for fruitful discussions. This work is financially supported by the Ministry of Science and Technology of China (2017YFA0302902 and 2017YFA0304600); the National Natural Science Foundation of China (grant No. 51788104, 11790311, 11922409, and 12004041) and the Beijing Advanced Innovation Center for Future Chips (ICFC). Work at Brookhaven is supported by the Office of Basic Energy Sciences, Division of Materials Sciences and Engineering, U.S. Department of Energy under Contract No. DE-SC0012704.

## Author contributions

M.L., D.Z., and Q.-K.X. initiated the project. M.L., Y.Z., and S.H. fabricated the samples and carried out transport measurements. R.Z., J.S., and G.G. grew the single crystals. M.L. and D.Z. analyzed the data and wrote the paper.

## Competing interests

The authors declare no competing interests.
