## [Peer Review File · Nature Communications]

REVIEWER COMMENTS

Reviewer #1 (Remarks to the Author):

The authors present magneto-resistance measurements on a number of thin and gated lightly doped BSSCO samples. They observe structure in the magneto-resistance traces that appears to have a well defined period according to the FFT analysis. The period of the structure varies little over a number of samples. The structure appears consistently at the low temperature and low but finite resistance portions of the resistive transitions of these lightly doped samples. They interpret the behavior as arising from the spontaneous formation of an array structure in the samples with a period of about 50 nm. They suggest this array structure reflects the spontaneous development of charge ordering in the system.

These conclusions are suitably novel and surprising for publication in this Nature journal. They suggest a new technique for exploring such charge ordering effects in strongly correlated electron superconducting systems. The consistency of the results across samples provides confidence that they are not an artifact.

There are two matters that I feel the authors should address prior to accepting the manuscript. One is related to the quality of the samples and the second addresses the applicability of their model to the data.

1) The $R(T)$ in Fig. 1 show evidence of two phase behavior in the form of two separate drops in the resistance, one at higher temperature than the other. Is this two phase behavior? Does it occur in all of the samples?

2) Have the authors considered plotting $\Delta R/R_N$? Their theory suggests that it will be independent of doping.

Reviewer #2 (Remarks to the Author):

The manuscript by Liao et al. reports an experimental finding where magnetoresistance oscillation is observed in thin flakes of high T_c cuprate B-2212. These oscillations appear around but below the superconducting transition temperature and at magnetic fields below 10T. The authors attribute the effect to the Little-Parks effect although the devices have no multiple-connected geometry defined. They suggest that certain charge ordering effect creates pockets of normal states with a characteristic length of 50 nm thus responsible for the Little-Parks oscillations. I found the result highly intriguing but there are several key issues for the authors to address more clearly.

My major concern is the attribution to the magnetoresistance oscillation to the Little-Parks effect. Conventionally the LP effect is observed in a defined doubly-connected geometry where the oscillation period matches with the device geometry. Conversely when the period matches it is safe to attribute the MR oscillation to the LP effect. In this case however, the sample is a continuous flake specimen, therefore there is no straightforward way to confirm the origin. Can we really exclude other quantum oscillations of normal states? I think this is a crucial part that the authors need to address with a better structured discussion than what was presented in the manuscript. The authors did point to a similar effect in TaS₂ (Ref. 11). There, the speculated LP effect appears more convincing as a clear H_{c2} transition can be observed, and the periodic oscillation appears more regular comparing to this work. I understand the difficulty in determining H_{c2} in cuprates but it nevertheless renders the interpretation of the results more difficult. Overall, the reasoning of attributing the oscillation to the Little-Parks effect is not convincing enough.

The authors are excluding Nernst effect reported in previous works citing different dependence on doping and magnetic field. I wonder why that is a stringent distinction to single out the effect observed here instead of the same effect with certain variations between samples. For instance, is

there any reason that Nernst effect should not be observed for $B > 6T$?

Is there any reason that this effect should only be observed in thin flakes as opposed to bulk samples?

The authors are presenting results obtained in several samples. To what extent are the conclusions representative? For example, can the doping dependence observed in S3 be confirmed in S1 as well?

Also it is not clear to me how the authors interpret the doping dependence (or independence for $p < 0.1$?) and what do they infer to the underlying mechanism.

I do not understand what the red highlight in Fig. 3 means. The caption suggests that they are "low-field resistance oscillations" but I think that term refers to the oscillation reported in this work, which is with a different period. The authors seem to suggest that the difference between the red highlighted periods and the experimentally observed periods (filled dots) in Bi2212 suggests the mechanism is not the conventional CDW but a new charge ordering effect. I do not understand the reasoning behind it.

Is the inferred 50 nm mesh size conclusive as which charge ordering it might be? For TaS2 it seems the length is at a comparable magnitude to this work but that was attributed to CDW.

As such, the authors should address the above listed issues before I could recommend the manuscript for publication.

Reviewer #1 (Remarks to the Author):

The authors present magneto-resistance measurements on a number of thin and gated lightly doped BSSCO samples. They observe structure in the magneto-resistance traces that appears to have a well defined period according the FFT analysis. The period of the structure varies little over a number of samples. The structure appears consistently at the low temperature and low but finite resistance portions of the resistive transitions of these lightly doped samples. They interpret the behavior as arising from the spontaneous formation of an array structure in the samples with a period of about 50 nm. They suggest this array structure reflects the spontaneous development of charge ordering in the system.

These conclusions are suitably novel and surprising for publication in this Nature journal. They suggest a new technique for exploring such charge ordering effects in strongly correlated electron superconducting systems. The consistency of the results across samples provides confidence that they are not an artifact.

There are two matters that I feel the authors should address prior to accepting the manuscript. One is related to the quality of the samples and the second addresses the applicability of their model to the data.

[REPLY] We thank the reviewer for the positive remark and the useful suggestion. We have added discussions on the two-phase like behavior as well as a figure showing $\Delta R/R_N$ in the revised manuscript. Details are discussed in the following points.

1) The $R(T)$ in Fig. 1 show evidence of two phase behavior in the form of two separate drops in the resistance, one at higher temperature than the other. Is this two phase behavior? Does it occur in all of the samples?

[REPLY] The reviewer is correct that the data in Fig. 1b show separate drops in the resistance in the superconducting transition, especially in the gated state of #1 to #4. In fact, all the lithium intercalated samples we studied in this work host certain kinks in the transition that seem to reflect multiple transitions (Fig. R1).

Fig. R1 Temperature dependent resistance curves of samples S1 to S4 at different states realized by lithium intercalation. Red curves indicate the states that show magneto-resistance oscillations. Black curves correspond to states that no oscillations are observed. The kink at 87 K probably stems from the region that remains at the initial doping level (nearly optimally doped).

However, we note that the multi-step transition does NOT correlate with the oscillations in magneto-resistances. We provide three pieces of evidence:

- (1) in the gated state of #1 to #3 of sample S1, there existed pronounced two-step transitions,

but we observed NO resistance oscillations.

(2) in the extremely underdoped regime where the two-step transition became hardly visible, we observed pronounced resistance oscillations (gated state of #5 for sample S1, as shown in Fig. 1c and Fig. R1; #6 and #7 for sample S4, as shown in Fig. R1 and Fig. R2 below).

Fig. R2 Data of sample S4. The magneto-resistance traces in the 6-th and 7-th gated states are shown in panel **b** and **c**.

(3) For the two samples shown in Fig. S3 and S4, under-doping was realized by the natural loss of oxygen. Although a kink at around 87 K can be seen, it clearly originates from the nearby regions that are still in the initially doping regime. The kink in Fig. S3b is from the thicker flake on the left side (see Fig. S3a). The kink in Fig. S4b is from the region outside the microbridge, which remains its initial doping since it is not exposed to the focused ion beam. In general, the underdoped region is well separated from the nearly optimally doped region in these two samples. Still, they show pronounced resistance oscillations.

We have added the corresponding discussions on the multiple transitions in the revised manuscript (paragraph before the subsection titled “doping dependence”).

Such a multi-step transition quite often occurs in high- T_c cuprate superconductors when its doping level is continuously tuned. Previous cases can be found in ionic liquid gated YBCO thin films both in the underdoped [PRL **107**, 027001 (2011): Fig. 1] and in the overdoped regime [PRL **108**, 067004 (2012): Fig. 1(b)] as well as a monolayer of BSCCO in the underdoped regime [Nature **575**, 156 (2019): Extended Data Fig. 4 d,g]. There are two possible origins. First, due to doping inhomogeneity, there emerge multiple regions with different doping levels such that their corresponding transition temperatures are different. Second, due to strong superconducting fluctuations, Cooper pairs form in spatially disconnected puddles at a relatively higher temperature and phase coherence across the puddles sets in at a lower temperature, thus giving rise to separate superconducting transitions [Nat. Commun. **9**, 4008 (2018)]. These two mechanisms may both contribute in experiments. However, the multi-step transition is not the focus of our present work and is unrelated to our key experimental findings.

2) Have the authors considered plotting $\Delta R/R_N$? Their theory suggests that it will be independent of doping.

[REPLY] Following the nice suggestion of the reviewer, we have added a new plot of $\Delta R/R_N$

as a function of temperature as Fig. 4d. It can be seen that by plotting the data this way, the peaks have similar heights. The remaining variation with doping mainly stems from the change in the energy quantities (E_1 and E_v).

Reviewer #2 (Remarks to the Author):

The manuscript by Liao et al. reports an experimental finding where magnetoresistance oscillation is observed in thin flakes of high T_c cuprate B-2212. These oscillations appear around but below the superconducting transition temperature and at magnetic fields below 10T. The authors attribute the effect to the Little-Parks effect although the devices have no multiple-connected geometry defined. They suggest that certain charge ordering effect creates pockets of normal states with a characteristic length of 50 nm thus responsible for the Little-Parks oscillations. I found the result highly intriguing but there are several key issues for the authors to address more clearly.

[REPLY] We thank the reviewer for the critical remarks, which help improve our manuscript significantly.

My major concern is the attribution to the magnetoresistance oscillation to the Little-Parks effect. Conventionally the LP effect is observed in a defined doubly-connected geometry where the oscillation period matches with the device geometry. Conversely when the period matches it is safe to attribute the MR oscillation to the LP effect. In this case however, the sample is a continuous flake specimen, therefore there is no straightforward way to confirm the origin. Can we really exclude other quantum oscillations of normal states? I think this is a crucial part that the authors need to address with a better structured discussion than what was presented in the manuscript. The authors did point to a similar effect in TaS₂ (Ref. 11). There, the speculated LP effect appears more convincing as a clear H_{c2} transition can be observed, and the periodic oscillation appears more regular comparing to this work. I understand the difficulty in determining H_{c2} in cuprates but it nevertheless renders the interpretation of the results more difficult. Overall, the reasoning of attributing the oscillation to the Little-Parks effect is not convincing enough.

[REPLY] We have substantially revised and reorganized our discussions to better distinguish our results from the other oscillations. In the first subsection of the “Discussion”, we present detailed reasons for excluding quantum oscillations of the normal state. In the second subsection, we significantly expand on the discussion of the Nernst effect, as we will explain in the next point. In the third subsection, we discuss the results of TiSe₂ in ref. 11 and draw clearer distinction.

In the revised manuscript, we present three arguments in the subsection titled “Quantum oscillations of the normal state?”: (1) the resistance oscillations that we observed always occur below B_{vs} —the magnetic field marking the boundary between the vortex solid and vortex liquid. Since B_{vs} is far below B_{c2} , the normal state is not reached when the resistance oscillations occur. We demarcate the $B_{c2} - T$ curve in Fig. 1d. (2) although we focus on the superconducting regime, we can also gate the sample further into the extremely underdoped situation ($p < 0.5$) such that the sample becomes non-superconducting. In the non-superconducting state, we observe no resistance oscillations (Fig. 3). (3) the mobility of our samples times the magnetic fields do not satisfy the criterion for observing quantum oscillations, i.e. $\mu B \ll 1$.

We wish to emphasize that although our observation is similar to that of Little-Parks effect on the qualitative level, a correct model has to take into account of the vortex dynamics (as illustrated

in the inset to the updated Fig. 4a) on the quantitative level. Such a quantitative analysis of the resistance oscillations was not pursued in the study of TiSe_2 (reference 11). By taking into account the vortex effect, we could extract the penetration depth in the underdoped regime. The values from different samples are consistent and they follow the expected trend as a function of doping. Based on the nice fitting and the sensible physical quantities extracted, we point out that “our speculation of a periodic structure captures important physics of the observed phenomenon.”

The authors are excluding Nernst effect reported in previous works citing different dependence on doping and magnetic field. I wonder why that is a stringent distinction to single out the effect observed here instead of the same effect with certain variations between samples. For instance, is there any reason that Nernst effect should not be observed for $B > 6\text{T}$?

[REPLY] The reviewer is correct that the difference in the observable magnetic field ranges is not a clear criterion. We have substantially rewritten the discussion on the Nernst effect. The revised texts are presented as the second paragraph of the subsection titled “Mesoscopic oscillations?”.

In the revised manuscript, we provide the physical reasons why the oscillations in Nernst signal were only observed at $B < 6\text{T}$. Essentially, the plastic deformation of vortex lattice no longer happens deep in the vortex liquid phase.

We provide two reasons against attributing our oscillations to the plastic deformation of vortex lattice. First, an in-plane magnetic field can drastically change the energetics of vortices via the addition of Josephson vortices. Therefore, the oscillations in Nernst effect, as ascribed to the vortices, should be sensitive to the in-plane magnetic field. By contrast, our experiments show that the resistance oscillations are independent of the in-plane magnetic field. Secondly, the oscillating features in Nernst signal were observed at optimal doping but vanished in the underdoped regime. It indicates that the effect of plastic flow of vortices becomes suppressed with decreasing doping. By contrast, none of our samples at optimal doping showed resistance oscillations. Resistance oscillations only appear deep in the underdoped situation.

Is there any reason that this effect should only be observed in thin flakes as opposed to bulk samples?

[REPLY] Based on our experiments, we believe that the resistance oscillations we observed are not limited to thin flakes. In fact, except for the 1-UC sample presented in the supplementary information, other samples we studied have thicknesses of about 50 nm (17 UC). These samples are therefore not in the two-dimensional limit and the transport results from them are representative of bulk properties.

The issue with the bulk samples (>100 layers) is that it becomes difficult to modulate the doping level from the optimally doped state to the extremely underdoped regime.

We have added the following texts in the subsection titled “Little-Parks oscillations”:
“The periodic structures are very similar across samples with different thicknesses and sizes, reaffirming an intrinsic origin. It also indicates that this effect is essentially representative of bulk crystals, although technically thin flakes can be more easily tuned in the doping level.”

The authors are presenting results obtained in several samples. To what extent are the conclusions representative? For example, can the doping dependence observed in S3 be confirmed in S1 as well? Also it is not clear to me how the authors interpret the doping dependence (or independence for $p < 0.1$?) and what do they infer to the underlying mechanism.

[REPLY] We address this point with the following revisions:

1. in the revised Fig. 3, we show the combination of temperature points and doping levels where the magnetotransport measurements were carried out. If resistance oscillations were observed, we use the filled symbols. Otherwise, we use empty symbols. By doing so, we show that different samples reproducibly show resistance oscillations in the lightly doped regime.

2. In the revised Fig. 4, we show the spatial periods extracted from all samples. They show almost the same values despite the differences in thicknesses and sample sizes.

3. We further show that the penetration depths from all samples by fitting the temperature dependent oscillating amplitude follow consistently the same trend. The doping dependence of the extracted penetration depth is in agreement with the expected doping dependence of the superfluid density.

Based on the above observations, we can conclude that our results reveal an intrinsic quantum phenomenon.

I do not understand what the red highlight in Fig. 3 means. The caption suggests that they are “low-field resistance oscillations” but I think that term refers to the oscillation reported in this work, which is with a different period. The authors seem to suggest that the difference between the red highlighted periods and the experimentally observed periods (filled dots) in Bi2212 suggests the mechanism is not the conventional CDW but a new charge ordering effect. I do not understand the reasoning behind it. Is the inferred 50 nm mesh size conclusive as which charge ordering it might be? For TaS2 it seems the length is at a comparable magnitude to this work but that was attributed to CDW.

[REPLY] For better clarity, we have replotted Fig. 3. The original figure contained two pieces of information: oscillation period (left vertical axis) and temperature (right vertical axis). We have moved the data about the oscillation period to Fig. 4. In the updated Fig. 3, we focus exclusively on the temperature and doping regime where our Little-Parks like oscillations occur.

We have also expanded on the discussion on the relation between the periodic structure we observed and the previously reported CDW in cuprate (the paragraph before the ending summary). Essentially, the CDW itself does not possess a period as large as 50 nm. In the case of TiSe_2 (ref. 11), the period inferred from the Little-Parks oscillations was even larger: 150-450 nm. There, it was proposed that the commensurate CDW forms periodic domains of a large size, and the incommensurate CDW situates in the domain boundary. In cuprates too, the previously observed charge order has a relatively small period of $4a_0$. In the discussion before the ending summary,

we have added our speculation that the large periodic structure may arise from the competition between two charge orders with small periods. In general, the large periodic structure is different from but intimately linked to the conventional CDW.

REVIEWERS' COMMENTS

Reviewer #1 (Remarks to the Author):

The authors have responded very constructively to the comments in my referee report. The discussion that they added to the text regarding the relation between evident multistep transitions and their observations of oscillations is clarifying. They have good arguments for why they are unrelated and the manuscript shows that they considered the possibility. And, their addition of $\Delta R/R_n$ curves in Fig. 4 lend direct support for their model of the phenomenon.

On the above basis and in accord with the comments in my earlier report I recommend publication of the manuscript in Nature Communications.

Reviewer #3 (Remarks to the Author):

The authors present in-plane magneto-resistance measurements of exfoliated Bi-2212 samples in low field (below 10T). Oscillations in the magneto-resistance appears few degrees below T_c in the highly under-doped region ($0.05 < p < 0.08$) and are absent at lower temperature and higher dopings. The oscillations appears to be controlled only by the out of plane part of the magnetic field, and FFT analysis of the oscillations reveals broad features around the same values (roughly 1 to 1.4 T⁻¹) in several samples. Careful measurements have been carried out to eliminate potential artifacts showing that these oscillations are reproducible.

In their analysis, the authors argue that these oscillations can be described by Little-Parks effect (dominated by vortex), having rejected other potential sources of oscillations. The authors suggest that interplay between multiple charge orders in the underdoped region could explain the appearance of large (50 nm) superstructures in the superconducting film.

The data appears solid and reproducible and the reported effect is intriguing and worthy of publication. The article is clear and well written. The analysis and conclusions are consistent.

I have however a small question/suggestion for the authors: as suggested by reviewer #1 the authors have checked that the fit of the variation with temperature of the amplitude of the oscillation scale with the normal state resistance for sample S3. I am curious whether a comparison can be made between different samples, especially between monolayer samples and thicker ones. From what I see in the color plot figures, the oscillations have more or less the same amplitude in all samples, but the monolayer sample should be a lot more resistive than a 50nm thick sample. Can a quantitative analysis be made by the authors?

Reviewer #1 (Remarks to the Author):

The authors have responded very constructively to the comments in my referee report. The discussion that they added to the text regarding the relation between evident multistep transitions and their observations of oscillations is clarifying. They have good arguments for why they are unrelated and the manuscript shows that they considered the possibility. And, their addition of Delta R/Rn curves in Fig. 4 lend direct support for their model of the phenomenon.

On the above basis and in accord with the comments in my earlier report I recommend publication of the manuscript in Nature Communications.

[REPLY] We thank the reviewer for recommending our paper for publication.

Reviewer #3 (Remarks to the Author):

The authors present in-plane magneto-resistance measurements of exfoliated Bi-2212 samples in low field (below 10T). Oscillations in the magneto-resistance appears few degrees below T_c in the highly under-doped region ($0.05 < p < 0.08$) and are absent at lower temperature and higher dopings. The oscillations appears to be controlled only by the out of plane part of the magnetic field, and FFT analysis of the oscillations reveals broad features around the same values (roughly 1 to 1.4 T⁻¹) in several samples. Careful measurements have been carried out to eliminate potential artifacts showing that these oscillations are reproducible.

In their analysis, the authors argue that these oscillations can be described by Little-Parks effect (dominated by vortex), having rejected other potential sources of oscillations. The authors suggest that interplay between multiple charge orders in the underdoped region could explain the appearance of large (50 nm) superstructures in the superconducting film.

The data appears solid and reproducible and the reported effect is intriguing and worthy of publication. The article is clear and well written. The analysis and conclusions are consistent.

[REPLY] We thank the reviewer for the very positive remark of our work.

I have however a small question/suggestion for the authors: as suggested by reviewer #1 the authors have checked that the fit of the variation with temperature of the amplitude of the oscillation scale with the normal state resistance for sample S3. I am curious whether a comparison can be made between different samples, especially between monolayer samples and thicker ones. From what I see in the color plot figures, the oscillations have more or less the same amplitude in all samples, but the monolayer sample should be a lot more resistive than a 50nm thick sample. Can a quantitative analysis be made by the authors?

[REPLY] We thank the reviewer for this nice suggestion. In the revised manuscript, we have updated Supplementary Figure 3 and employed sheet resistance instead of the previously used measured resistance. By doing so, the data of sample S1 in Fig. 1 can be compared with the data of sample S5 in Supplementary Fig. 3, because both are using the sheet resistance R_s . Indeed, the oscillation amplitude increases as the sample thickness reduces. We further remark that we use the directly measured resistance for sample S3 in Fig. 2. This sample has irregular contacts made of thin flake of graphite, as explained in the Method section. We emphasize that the variation in the resistance amplitude does not affect our quantitative analysis. Figure 4b includes the results from fitting for samples S1 to S5.

As an example, we show in Fig. R1 the oscillation amplitudes in sheet resistance for S4-#5 and S5. Their doping levels are estimated to be the same. Despite the sharp difference in the absolute values of the amplitudes, the temperature dependences can be well fitted by using Eq. (1). We show in Fig. R1 b and d the normalized resistances as well. These peaks are comparable to those shown in Fig. 4d. The remaining variation in peak height and temperature can be accounted for by the variation in the remaining fitting parameters, such as the London penetration depth (shown in the inset Fig. R1a and c). It indicates that the penetration depth may vary from sample to sample.

Fig. R1 Temperature dependences of the oscillation amplitude ΔR_s and the normalized ones for sample S4 (about 50 nm thick) and sample S5 (1-UC)